# Diagnostics of the RC Roofing Structure of the 100-Year-Old Municipal Theatre Facility

**DOI:** 10.3390/ma15217438

**Published:** 2022-10-23

**Authors:** Marta Kosior-Kazberuk, Janusz Ryszard Krentowski, Maciej Wardach

**Affiliations:** Faculty of Civil Engineering and Environmental Sciences, Bialystok University of Technology, Wiejska 45E, 15-351 Bialystok, Poland

**Keywords:** RC beams, NDT, historic structures, degradation, strengthening

## Abstract

Reinforced concrete has been a widely used material for the construction of buildings for many decades. However, with the passage of time, the material characteristics and connection of structural elements gradually degrade. Development in measurement technology makes it possible to efficiently obtain data on the current state of the structure and material characteristics using non-destructive methods, with limited or no destructive testing. The paper presents the analysis of the condition of the reinforced concrete roof of a 100-year-old theatre building in terms of its further use after planned modernisation. The tests carried out in situ as well as the computational analysis of structure are described. Based on the test results, the current load-bearing capacity was assessed and the limit state conditions were checked. Limitations on the accuracy of the non-destructive test results in relation to the destructive test results were formulated. Options for the strengthening reinforced concrete beams with regard to structural and technological considerations were analysed.

## 1. Introduction

Poor management and lack of maintenance of buildings that have been in continuous use for more than a century threaten the durability of the structure and, consequently, the safety of users [1,2]. Ignoring the need for retrofit work can accelerate the degradation processes of building materials [3]. External structural elements, exposed to the direct effects of the environmental influence of the weather, will not wear out quickly if the facility is diagnosed correctly, and the repairs carried out can effectively extend the service life [4,5]. Degradation processes are influenced by both environmental factors and the natural ‘ageing’ of materials. The errors occurring at the design and construction stage, or incorrect operation or condition assessment, are also factors that can generate hazardous situations [6,7,8,9]. The public buildings should be constantly monitored and should comply with the most restrictive standards. Properly carried-out renovation work allows them to operate safely and regain their architectural qualities [10,11].

Reliability of structures, i.e., the ability to meet the requirements of load-bearing capacity, serviceability and durability, is a fundamental design issue. Methodological principles for the design of structures in Europe are included in the standard [12], which is based on the method of limit states and partial factors. In the case of calculation of newly designed structures, the materials with strength parameters adequate for the load-carrying requirements placed on them are adopted. The problem occurs with existing structures, for which static calculations must be carried out that take into account the actual physical and strength properties of materials and state of degradation. Before proceeding with design work for the reconstruction, renovation or strengthening of such structures, it is necessary to carry out a number of tests of material parameters. To avoid design errors, it is reasonable to analyse the cases of structural failures that have occurred. These failures occur in both industrial [13,14,15] and civil structures [16]. Analysis of the origin of unintentional structural failures provides a valuable testing ground for scientists and designers. The combination of advanced research methods together with numerical calculations allows the causes of loss of load-bearing capacity to be identified, which contributes to improving design work and avoiding the duplication of errors.

Public buildings constructed in the 20th century were usually constructed as masonry. Widely known ways of maintaining, repairing or strengthening such structures have been described in the literature [17,18]. However, between the wars, i.e., in the 1930s, reinforced concrete structures and mixed or combined structures such as reinforced concrete and masonry or reinforced concrete and steel structures were also built. These structures were usually characterised by a long service life, but nowadays, due to their service life of about a century, they require immediate repairs or complex modernisations. A change in the function of or way of using the building requires a detailed analysis of the conditions of the elements of historic structure and the properties of materials used over many years.

The development of measurement technologies is positively influencing the diagnosis of building structures. Modern testing methods such as digital image correlation [19,20,21], as well as laser scanning [22,23] and ultrasonic methods [24,25], allow for non-invasive monitoring of structures. Non-destructive testing of concrete elements has been popularised in [26,27,28]. Due to the exposure of the elements to aggressive industrial influences, chemical tests of the concrete also provide reliable results [29]. The extensive descriptions of experimental methods of detecting structural damage that use ultrasonic methods based on the nonlinear Lambda wave principle were presented in [30] and the methods based on colinear nonlinear mixed-frequency ultrasound can be found in [31]. The systematisation and summarisation of the latest methodologies and technologies for vibration-based structural health monitoring can be found in [32].

As a result of the conducted research, it becomes possible to carry out an analysis of the current technical state of existing structural and architectural solutions. The results of research also make it possible to carry out calculations that allow for an unambiguous assessment of the structure’s load-bearing capacity [33]. Strengthened structural components are also being experimentally studied in research labs [34,35], contributing to the development of repair methods. In the case of complex modernisation of the building, all structural elements should comply with the limit state conditions specified in valid standards and regulations.

## 2. Historic Theatre Roof Structure

The described diagnostic measures are illustrated with an example of the research procedure and concepts for reinforcing the RC roof of historic theatre, located in north-eastern Poland. The main part of the theatre was built between 1933 and 1938. After nearly 100 years of exploitation, the city authorities decided that the building, which is listed as an historical monument, needed to be renovated. The aim of the planned work was to bring the facility up to current standards in terms of stage technology and acoustic requirements, and, above all, structural solutions to ensure the safety of users.

The load-bearing structure of the building consists of solid ceramic brick walls on lime mortar, based on concrete foundation walls. The inter-storey floors were built as reinforced concrete slabs supported on reinforced concrete beams and external walls. The roof slab was made as a monolithic reinforced concrete structure. The main structural elements of the roof are the reinforced concrete beams, varying in height from 1.00 m to 1.30 m, on which the box section roof is supported. The beams, with a clear span of 15.74 m, are supported on reinforced concrete columns with a cross-section of 0.50 × 0.70 m. Bevels were made in the support zones of the beams to stiffen the joints. The schematic view of the beams on the theatre plan is shown in Figure 1a. The structural cross-sections are presented in Figure 1b–d.

The box section ceiling structure was made as a monolithic one (Figure 2b). The dimensions of the ceiling are shown in Figure 1e. In the central part of the ceiling, between two central beams, there is a skylight of a width corresponding to the ceiling span between the beams. The main structural elements of the ceiling in this zone are the ribs (Figure 2a) with cross section 0.30 × 0.12 m. The axial spacing of the ribs is approximately 0.90 m. The ceiling is insulated with hard mineral wool and covered with two layers of roofing felt.

A steel-framed technological platform was suspended to the beams above the stage (Figure 3), providing additional load to the existing structure. The main structural elements are steel hangers, to which beams made of 180 mm high rolled I-beams are attached. C100 joists (Figure 3a) are supported on the beams, with steel deck gratings made of flat bars. Lighting bridges, winches and other equipment necessary for the theatre’s stage are mounted on the technical platform.

Due to the planned change, i.e., increased loads resulting from the necessity to use new technology, as well as new acoustic requirements, a variant analysis of how to adapt the structure to current regulations became necessary. Whole equipment comprising the so-called stage technology was designed to be suspended from the reinforced concrete roof. A similar solution was applied in the theatre building to that time. An important design aspect is also the current fire protection standards, which are more stringent than years ago. The main load-bearing elements, especially the beams, are vulnerable to damage during a fire. Elements weakened by damage to the cover layer and corrosion of the reinforcing bars are less able to withstand the temperature of a fire, which can consequently lead to the disaster of the whole structure [36]. Analysis of the real state of the structure, additionally taking into account the thermal effects on the state of damaged structural elements, recommends that specialised calculation models are used [37,38,39]. The effectiveness of the obtained computational results is determined by the quality of the diagnostic data entered into the calculation programs.

## 3. Test Procedures for Reinforced Concrete Beams

To determine the actual strength parameters, as well as the progress of the degradation processes of the structural elements, the series of in situ tests were carried out. Non-destructive (Figure 4) and visual methods of concrete and steel testing were used, and local excavations were made. Due to lack of knowledge regarding the condition of the structure at current stage of the study, it was decided not to take specimens for destructive testing of concrete and steel. The results of non-destructive testing are subjected to a higher risk of error in comparison to the results obtained by destructive methods. However, in many cases, they are the only source of data allowing a preliminary assessment of the technical condition of structure.

Visual examination revealed that repair work to the beams consisting of reprofiling was conducted in the past (Figure 5a). The reprofiling had been carried out incorrectly, as the concrete cover of the reinforcing bars was found to be missing on significant areas of beam surface. In addition, the repair mortar easily peeled off after a light impact. This means that the protective layer did not achieve proper adhesion with the structural concrete. Numerous irregular cracks, with an opening width that did not exceed 0.5 mm, were found on the vertical surfaces of beams (Figure 5b,c).

After removing the roughcast and outer layer of concrete, it was possible to determine that the cracks in the area of the opencast were superficial. No excessive cracks were found in the span zone or in the support zone as a result of exceeding the bending moment limits or the shear force. Potential scratches in these zones were covered up by the applied cement mortar repair treatments. No excessive scratches were found on the reprofiled lower surfaces of beams. The roof surface was airtight, and no signs of streaking were observed. The test of moisture in the structural materials, carried out using a digital moisture meter (Figure 4c), gave the results ranging from 45 to 60% RH.

To estimate the compressive strength and homogeneity of the concrete of several structural elements, a sclerometric method was used. On both sides of all beams, 5 surfaces each were prepared with power tools to read the number of rebounds. On this basis, the strength of concrete corresponding to contemporary C16/20 class was estimated, noting that the concrete is strongly heterogeneous. Nowadays, concrete of this class is not applied for the main structural elements, but only for secondary elements or base layers. A dull sound was found locally, which is characteristic of delaminated elements with voids and discontinuities inside the concrete section. The varying rebound number can also indicate carbonisation of the concrete, which leads to hardening of its surface layer. The process of carbonisation is associated with the loss of passivation of reinforcing steel, which can result in the loss of adhesion of the reinforcement and its corrosion.

In order to confirm the results of the sclerometric tests, the ultrasonic tests were carried out. Two-sided access to the component using longitudinal wave transducers was used (Figure 4a). Six measurements were taken on each beam—2 readings in each of the support and the span zones. Regardless of the location of the transducers, it was not possible to carry out a measurement due to the high interference and wave dispersion, confirming the strong degradation of the reinforced concrete elements. Locally, a surface wave velocity of about 2200 m/s was measured, which, according to [40], qualifies the concrete as weak. Next, measurements were conducted using transverse waves with a Pulse-Echo transducer (Figure 4b). Readings were taken for both sides of the beams where measurements were made with longitudinal wave transducers. The ultrasonic wave velocity was extremely low and averaged 1100 m/s, where the typical value for concrete is 2000–2200 m/s. In some places, the measures were impossible. The results confirmed the hypothesis of discontinuity of the medium due to internal defects in the concrete structure. Plots of longitudinal wave propagation (measured with single-sided access) and transverse wave propagation (using the Pulse-Echo head) are presented in Figure 6.

Electromagnetic scanning of the beams was carried out to determine the quality of the reinforcement work (Figure 7). The measurements made it possible to assess the distribution of the reinforcement, which was then implemented in the static calculations. Insufficient cover thicknesses were locally found. Initial corrosion processes of the steel reinforcement were inventoried at these locations. The too-low cover thickness or local cover losses have a negative effect on the structure’s resistance to fire temperatures.

The diameter and distribution of the reinforcement were determined on the examination of excavations, which were carried out on the beam surface and the analysis of areas without concrete cover (Figure 8). The number of excavations was limited due to concerns about the poor technical condition of beams.

The research allowed it to be established that the bottom reinforcement of the beam was made up of 5 Ø32 mm diameter bars of non-ribbed steel. The transverse reinforcement was made of four-cut stirrups of Ø12 mm bars of non-ribbed steel. In order to simplify the implementation of the scan results into the numerical model, the distances between the stirrups were averaged due to the very uneven spacing of the stirrups. At the supports, the average spacing of the stirrups was assumed to be every 5 cm, further on every 12 cm and in the middle of the span every 20 cm. The box ceiling ribs were reinforced with 2 Ø12 bars at the bottom. The stirrups were made as double-cut Ø8 mm bars and were spaced at 15 cm and 20 cm intervals. Based on archival data from buildings constructed at the same time and constructed in the same area as the building under investigation, the yield strength of the reinforcement bars was assumed to be 230 MPa. Details of the beam reinforcement are shown in Figure 1c–e.

## 4. Computational Verification of Stress State

The calculations were performed in ANSYS [41], assuming the stress–strain relationships of concrete and steel according to EC2 [42] Section 3.1.7 and Section 3.2.4. The FEM mesh was assumed to be rectangular in shape, where the maximum side dimension does not exceed 250 mm. The grid size results from the most favourable calculation time with a satisfactory level of result accuracy. Beams and columns were modelled with SOLID186 elements, i.e., solid elements defined by 20 nodes with three degrees of freedom per node. The element is characterised by ductility, high deflection and large deformation capacity. Reinforcing bars were modelled as reinforcement of the solid elements using REINF264.

On the basis of sclerometric tests, the strength of the concrete was estimated to be equivalent to contemporary class C16/20 (f_ck_ = 16 MPa, f_cd_ = 11.43 MPa), and the yield strength of the reinforcing steel was assumed to be f_yk_ = 230 MPa (f_yd_ = 200 MPa). The number and spacing of bars were assumed on the basis of the inventory, the excavations and the results of the electromagnetic tests. Static calculations for the design values of the strength parameters of concrete and steel were carried out according to [42]. The effective width of the box section floor, representing the top shelf of the beam, was also taken into collaboration. The calculation of the effective width of the *b_eff_* was carried out according to [42].

The calculations were carried out for two static schemes, i.e., a frame scheme with the beams rigidly connected to the columns and an articulated scheme with the simply supported beams. The plasticisation of the nodes can lead to articulation and hence a change in the static scheme from a frame to a simply supported beam. In view of the lack of reliable data in terms of the formation of frame joints, the uncertain quality of the concrete and steel, and the expected service life over the next few decades, it is reasonable to carry out calculations for both variants.

The beams were loaded with dead loads of the roof (g = 5.6 kN/m^2^), as well as climatic loads, i.e., wind (w = 0.35 kN/m^2^) and snow (s = 1.28 kN/m^2^), and service loads (q = 2.0 kN/m^2^). Loads were assumed in accordance with standards [43,44,45], while calculation combinations were formulated in accordance with [12]. The results of the calculation are shown in Figure 9. The programme automatically takes into account the dead weight of the beam, which is approximately 14.38 kN/m and represents about 40% of the total dead loads.

On the basis of the calculations, it was indicated that the ultimate limit state and serviceability limit state would not be exceeded for the beams rigidly connected to the columns. In the case of the simply supported beam scheme, the stresses of 202.6 MPa in the reinforcing steel exceed the normal values of 200 MPa and the ultimate limit state is not fulfilled by only 1%. It is worth noting that in both static schemes the serviceability limit state due to deflection is fulfilled.

Especially dangerous for the operation of the beams will be a change in the static scheme with plasticisation of the joints connecting the columns with the beam. This situation will result in a redistribution of forces in the beam and an increase in the bending moment in the span zone. In this situation, the bending capacity of the beam will be exceeded, which poses a safety risk to the structure and the users of the building. This phenomenon may occur, for example, in the case of locally lower strength parameters of materials used to construct the beams compared to the assumed design material features. These lower material parameters may by caused by poorly compacted concrete mix resulting in concrete heterogeneity, delamination and cracking.

However, it should be noted that the static-strength analysis was carried out without taking into account reductions in the load-bearing capacity of the beams caused by internal defects such as possible poor compaction of the concrete mix, delaminations, concrete cracking and ageing. As a result, there is a real risk of a higher strain on the elements due to the very poor quality of the concrete and the reinforcement work. The calculations carried out for the two static schemes showed that in the case of plasticisation of the joints, the ultimate limit state would be exceeded, even without taking into account internal defects. Therefore, in the case under analysis, consideration of the decrease in bearing capacity caused by such degradation would be negligible.

## 5. Analysis of Opportunities for Strengthening

With regard to the risk of exceeding the ultimate limit state and serviceability limit state as a result of new designed loads, it is necessary to select an economically and constructionally optimal design solution. The problem of strengthening reinforced concrete beams has been analysed many times in available monographs and scientific articles [46,47,48,49]. Based on published solutions and their own research experience, the authors have made a variant assessment of the possibility of reinforcing the existing structure in terms of its continued safe use.

### 5.1. Concretising the Existing Structure

One option for increasing the load-bearing capacity is to concretise the structural element [50], i.e., to increase its dimensions, above all, the useful depth of reinforced concrete section. This method simultaneously rebuilds the lost cover layer. During the selection of the sprayed mix, the factors of environmental aggression and the expected strength parameters have to be considered. In addition to the parameters of sprayed concrete, the quality of the strengthening is also affected by the parameters of shotcrete and the type of machinery used [51]. These factors generate a problem in determining the parameters of the applied concrete. The literature [52,53] presents the results of experimental studies related to the determination of Young’s modulus and shear capacity. Thus, this method is relatively simple in terms of execution, but the reinforcement parameters are very difficult to determine precisely. In the case of the analysed beams, for which multiple reprofiling had previously been used, concretising the element would have been troublesome due to the variable adhesion of the individual concrete layers, i.e., the original and after reprofiling. The reinforced concrete must also have an adequate compressive strength, which is very low in the case of the beams analysed. The application of such a method requires the removal of the carbonatised concrete layer up to the surface of the material with correct elastic and strength characteristics. On the basis of preliminary destructive tests, it was determined that it would be necessary to remove the concrete to a depth of several centimetres around the entire circumference of the existing beams. The problem of connecting the reinforcing layer to the existing uneven-surfaced structure could be solved by using additional anchors made of ribbed steel bars, set to a depth of several dozens of centimetres, through high-strength mortar.

The authors also considered the option of surrounding the cleaned surfaces of beams with 3 mm diameter bar nets, fixed with bars embedded permanently in the original structure. A few centimetres’ layer of sprayed concrete could be effectively placed on such prepared surfaces. The similar solution was described in [54], where the supporting structure of industrial tanks was reinforced.

### 5.2. Demolition of the Roof Structure

The possible option is to remove the slab of box section roof while leaving the beams. In this case the beams relief will be provided but, at the same time, the beams will lose their plate bracing. The partial reduction in load bearing capacity can be expected as the top shelf of the T-section of beam will be removed. Between the beams, lightweight steel trusses can be constructed on which the new roof, made of structural steel plate and a purlin system of cold-formed steel sections, would be supported. The steel trusses would be supported on steel replacements articulated on the existing columns. This way, it would be possible to relieve the load off the existing beams and to transfer all the loads to the new structure. The beams, carrying only their own weight in this option, could be used to carry process loads, e.g., from light installations.

The most time-consuming and expensive solution is to remove the entire reinforced concrete ceiling with beams and build a new roof in lightweight steel construction with trapezoidal sheet metal cladding. However, acoustic considerations are a contraindication to the implementation of a steel-framed roof. Due to the low weight of such a structure and the specific characteristics of the steel elements, additional screens will be required to dampen vibrations caused by acoustic waves. In case of a decision to demolish the beams, the most favourable solution in terms of lead time would be to use prefabricated prestressed beams. Once the existing elements had been cut out, the columns would need to be profiled to accommodate the new reinforced concrete girders.

It should be noted that the number of box ceiling tests are also required for both the above solutions. Aging processes and the influence of the external environment (the roof is most exposed to water and temperature changes) may have reduced its load-bearing capacity, just as in the case of beams. Determining the degradation state of the roof should precede design work for demolition. It is important to prevent structural failure also in the last stage of the structure’s life, i.e., its dismantling. Improperly planned demolition work can lead to a structural disaster and risk the lives of workers.

### 5.3. Other Strengthening Concepts

Another strengthening option would be to place steel trusses at mid-span of the beams, without removing the box section roof. The trusses would be supported on the columns by steel or reinforced concrete replacements. In this way, the beams would be relieved of half the applied loads. The risk is the change in the static scheme of the floor slab resulting from the addition of more supports. The execution problem is the need to weld a steel structure of low stiffness to a reinforced concrete section in order to get the existing floor to cooperate with the new truss.

Strengthening by means of reinforcement with composite tapes is often used to strengthen reinforced concrete beams [47,55]. Such strengthened structural elements are characterised by various types of damage, which are widely described in [49]. Predictive models for typical damage are also known [56]. The creep of the concrete and the type of adhesion are important in terms of the reliability of such a strengthening [57]. Various configurations of strengthening reinforced concrete beams with composites were experimentally tested in [58]. The commonly used methods of increasing the load capacity can be divided into passive and active ones. The passive method consists of passively reinforcing the tension zone of the component with tapes. The composite tape is active only when the deformation of the structure has increased. In the case of beams considered, this method would be ineffective, as the dominant load is the dead weight of the roof and beams. The active method relies on the external pre-stressing of the structure by means of laminations using tension devices attached to the members to be strengthened. The undoubted advantage of this method is that the structure does not need to be unloaded but the problem is the making of the anchor block fixings. Due to uncertain quality and strong heterogeneity of the concrete, there is a risk that the attachment of the blocks to the reinforced concrete beams will not be carried out correctly. In addition, due to the close spacing of the reinforcing bars, it is impossible to fix the anchor bars without damaging the reinforcement. For both methods, restrictive fire regulations are also a contraindication to their application. The elements would have to be secured in accordance with current standards, which would entail additional costs as well as additional incremental loads.

Nowadays, with the development of construction chemistry products, many modern, specialised mortars are used for dry or wet spray application. The papers [59,60] present the results of experimental studies of strengthening using repair mortars. The mixtures are characterised by the possibility of application in any exposure class of structural elements. Properly selected mixtures can be used even in the most aggressive chloride ion environment, i.e., in coastal engineering [61]. The considered beams had previously been reprofiled, but the repair work was carried out incorrectly. The placement of another spray layer is considered to be a technically incorrect solution, which will not fulfil the expected strengthening effect due to the lack of cooperation with the original reinforced concrete core and will not provide the required load-bearing capacity for the newly designed loads.

## 6. Discussion

Historic structures in operation should be strengthened in such a way that changes occurring during the design life, taking into account environmental influences and the expected level of maintenance, do not reduce the performance of the structure below the intended level. The design life of the theater’s continued use, according to [12], is 100 years. However, the poor quality of concrete and reinforcement work, as well as numerous defects in the structure of the concrete and inadequate reprofiling, indicate the unquantifiable risk of using the beams for further service [8].

The observed cracks on the lateral surfaces of the beams are mainly caused by concrete shrinkage. The size of the crack is significantly influenced by the large thickness of the reinforcement lagging, as well as the insufficient amount and spacing of the horizontal shrinkage reinforcement. Cracks of this type do not threaten the safety of the structure but affect the homogeneity of concrete.

The properties of steel bars produced in the early 20th century may now differ significantly from their original properties, as they have been subject to an ageing process. As steel ages, its tensile strength, yield point and hardness change. The loss of steel’s ductile properties can result in unsignalised damage or failure of the beams as a result of exceeding the yield strength. It should be taken into account that in the analysed beams, the ageing process of the steel will continue to progress. Heterogeneity and progressive carbonation of concrete can negatively affect the properties in terms of protection of reinforcing bars against corrosion. Progressive corrosion, on the other hand, is associated with the loss of bar cross-sections, which corresponds to a reduction in load-bearing capacity.

The listed defects may cause a local reduction in the strength parameters of the materials resulting in the risk of exceeding the ultimate limit state. The formation of articulations in the connection zones between columns and beams will cause an increase in the bending moment in the beam spans, resulting in a risk of failure or structural catastrophe.

## 7. Conclusions

The research on the historic ceiling and the discussion of reinforcement options allowed for the formulation of the following conclusions:in the case of historic unit objects, limited access to structural elements and the inability to conduct full destructive testing force the use of a combination of non-destructive testing methods, without significant interference with the historic building;a properly selected combination of non-destructive testing methods allowed for a comprehensive assessment of the technical condition of the facility, necessary to develop variants of its strengthening;in the case that we are faced with the dilemma of how to preserve a monument in accordance with the conservation doctrine and at the same time secure it in such a way that it meets the requirements of the safe stay of people, it is necessary to analyse the options for repairing the object;in the case of historic buildings, the amount of questionable information and assumptions is so large that both the survey and numerical calculations only estimate the results. Therefore, the experience of a construction appraiser who makes proper design decisions is necessary. According to the principle of “Practice Makes Perfect,” the greater the experience of the appraiser, the greater the certainty regarding the obtained results of research and numerical analysis. This affects the effectiveness in making decisions on the continued exploitation of the building.

## Figures and Tables

**Figure 1 materials-15-07438-f001:**
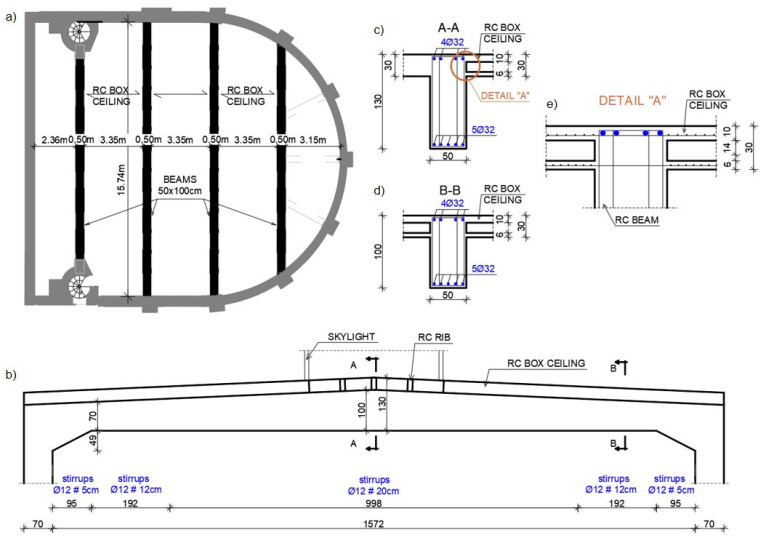
The layout of the roof ceiling: (**a**) plan of the beams; (**b**) view of the beam; (**c**,**d**) structural cross-sections of the beam; (**e**) layout of the box ceiling.

**Figure 2 materials-15-07438-f002:**
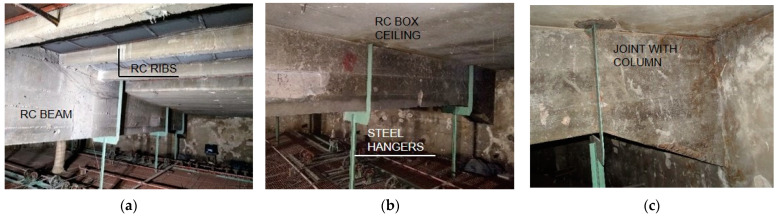
View of the beams: (**a**) at the skylight location; (**b**) near the column support; (**c**) connection to the column.

**Figure 3 materials-15-07438-f003:**
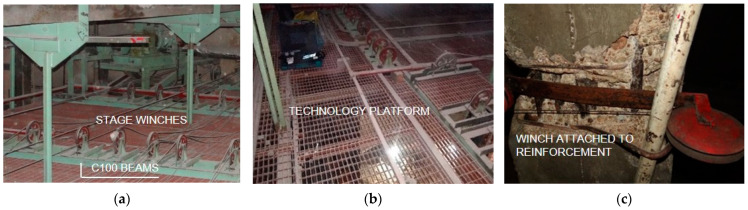
View of the technological platform: (**a**) winches: (**b**) the platform grid: (**c**) elements attached to the lower reinforcement of the beams.

**Figure 4 materials-15-07438-f004:**
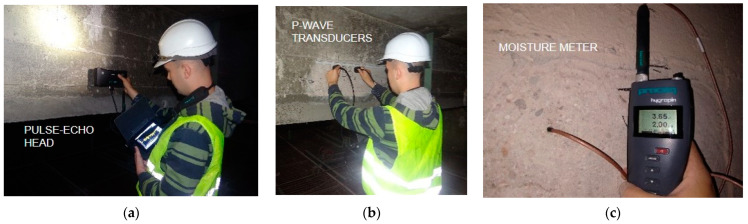
Non-destructive testing: (**a**) Pulse-Echo method; (**b**) ultrasonic testing using longitudinal wave transducers: (**c**) concrete moisture testing.

**Figure 5 materials-15-07438-f005:**
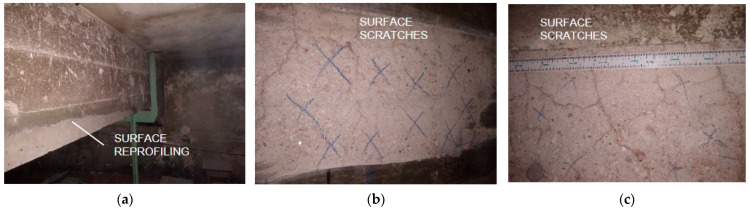
Visual tests: (**a**) view of reprofiled surfaces; (**b**,**c**) surface cracks.

**Figure 6 materials-15-07438-f006:**
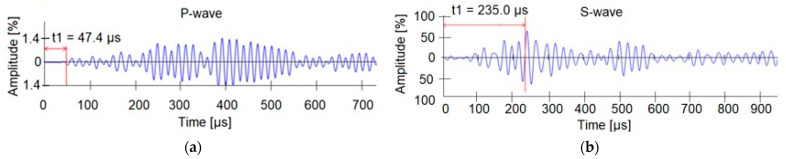
Wave propagation diagram: (**a**) longitudinal wave; (**b**) transverse wave.

**Figure 7 materials-15-07438-f007:**
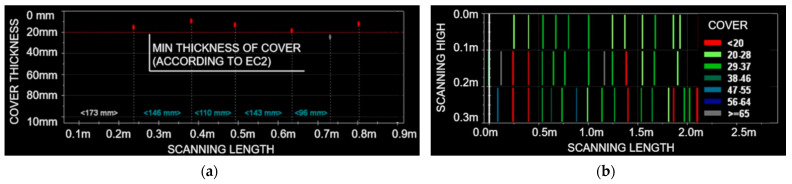
Results of electromagnetic scans: (**a**) linear scan; (**b**) multilinear scan.

**Figure 8 materials-15-07438-f008:**
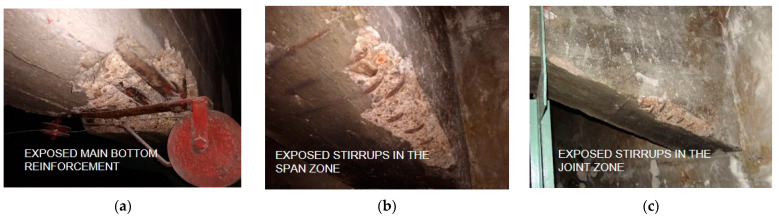
Excavations on beam’s surface: (**a**) at the point where the main reinforcement bars are connected to the winch; (**b**) at the point of insufficient cover in the beam span; (**c**) at the point of insufficient cover in the bevel.

**Figure 9 materials-15-07438-f009:**
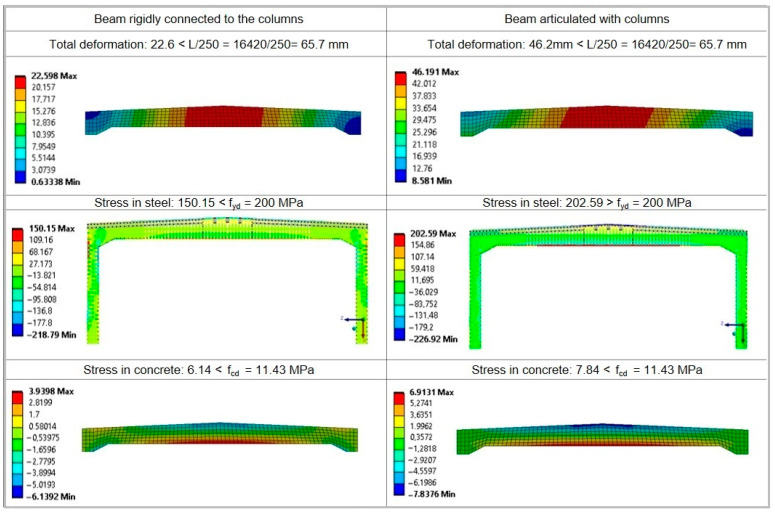
Static calculation results.

## Data Availability

The data presented in this study are available on request from the corresponding author.

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
