# Peer review of "Diagnostics of the RC Roofing Structure of the 100-Year-Old Municipal Theatre Facility"

_materials, 2022, doi:10.3390/ma15217438_

Round 1
Reviewer 1 Report
The manuscript "Diagnostics of the RC roofing structure of the 100-year-old municipal theatre facility" deals with some interesting analyses on the condition of the reinforced concrete roof of a 100-year-old theatre building in terms of its further use after the planned modernization. The tests carried out in-situ as well as the computational analysis of structure were described. Based on the test results, the current load-bearing capacity was assessed and the limit state conditions were checked. The authors have very nicely represented the work with interesting data. The analysis results are very interesting indeed and are backed with simulation results. I strongly recommend this manuscript for publication. The manuscript, however, requires some revisions before it can be considered for publication.
1. The introduction on similar work is very limited and does not cover similar experiences on the topic. I strongly recommend authors give a broader overview of similar works on the topic. I suggest authors consult following manuscripts in their revision:
- Hanxin Chen, Mingming Liu, Yongting Chen, Shaoyi Li, Yuzhuo Miao, "Nonlinear Lamb Wave for Structural Incipient Defect Detection with Sequential Probabilistic Ratio Test", Security and Communication Networks, vol. 2022, Article ID 9851533, 12 pages, 2022. https://doi.org/10.1155/2022/9851533
- Chen H, Li S. Collinear Nonlinear Mixed-Frequency Ultrasound with FEM and Experimental Method for Structural Health Prognosis. Processes. 2022; 10(4):656. https://doi.org/10.3390/pr10040656
- Zhang, C., Mousavi, A. A., Masri, S. F., Gholipour, G., Yan, K., Li, X. (2022). Vibration feature extraction using signal processing techniques for structural health monitoring: A review. Mechanical systems and signal processing, 177. doi: 10.1016/j.ymssp.2022.109175
- Huang, H., Guo, M., Zhang, W., & Huang, M. (2022). Seismic Behavior of Strengthened RC Columns under Combined Loadings. Journal of Bridge Engineering, 27(6). doi: 10.1061/(ASCE)BE.1943-5592.0001871
2. This manuscript is all about failures and reliability. Yet, these two concepts are hardly addressed in the introduction part. Instead, authors have devoted first few sentences to talk about art and human civilization. I suggest authors consult papers, related to failure analysis and reliability.
3. Please provide more specific details in the conclusions. The current conclusion is rather generic. I suggest you re-write conclusions in bullet-point.
4. A revision in English is recommended, though overall English is acceptable.
5. The discussion is rather basic and short. Consider merging it with results to have a unified “results and discussion” part.
Author Response
Dear Reviewer,
We are very grateful for your careful reading of our manuscript. We appreciate the suggestions, which we have carefully addressed below (the reviewer's comments are italicized). We believe that our manuscript has been greatly improved thanks to all comments. We hope that our text after correction, meets the criteria necessary for publication in the journal Materials.

Reviewer 2 Report
This paper analyzes the roof of a century-old reinforced concrete theater. In order to ensure the safety of continued use, a series of experiments were carried out on the theater, the bearing capacity analysis was carried out, and the solution was finally given. Although a lot of work has been done, the article describes the structural testing process of the house with a lot of space, and the diagnostic analysis part is not detailed enough. Therefore, a major revision is needed to make the research of deep significance. The following questions also need to be improved:
1. A chapter should be added to introduce the various detection technologies widely used at this stage and make appropriate analysis and comparison;
2. The third section of the article describes the surface condition of the structural member, but it is not enough to describe the crack state. Please add other aspects appropriately, such as the detachment of the component protective layer, the corrosion of the member, etc.
3. The article has repeatedly mentioned that the fallout of the component protective layer has a negative impact on the structure, so what is the solution?
4. The fourth section of the article analyzes the static strength of the structure, but does not consider the reduction of the bearing capacity caused by the internal defects of the component. Is this analysis meaningful?
5. Section 5.2 introduces one of the solutions "disassembly roof structure". Considering the long service time of the building, there are risks such as aging, and whether the interference of the demolition process will cause potential safety hazards, please analyze it in detail.
6. The introduction should focus on the content related to the topic of the article.
7. The article has emphasized the heterogeneity of concrete many times, which makes its quality and compressive strength uncertain. So how to ensure the scientificity of the test results?
Author Response

(The authors gave the same response as above.)

Reviewer 3 Report
This paper presented the analysis on the condition of the reinforced concrete roof of an ancient theatre building through in-situ monitoring test and numerical modelling. Although the research is interesting and of engineering significance, the paper is not recommended for publication as it is. The detailed comments are as follows:
1. The current issues and related studies are not summarized sufficiently. It should be improved and pointed.
2. The description and data analysis of the testing is not detailed, especially some technical information.
3. The figure to present the bar distributed in beams is lacked.
4. The strengthening part is too general, and needs to be improved by more data analysis and discussion.
5. In conclusions, the useful data is not provided and the words prove general and lack of academic contributions.
Author Response

(The authors gave the same response as above.)

Reviewer 4 Report
This manuscript studied the issue of “Diagnostics of the RC roofing structure of the 100-year-old municipal theatre facility”.
First of all, I would like to thank the authors of this manuscript for the effort they put into making it. The paper needs to be rewritten, its objectives well redefined, and details of the experimental program clearly stated. On the other hand, I have added some comments with the main objective of improving the manuscript.
i. What is the innovation point or significance of the study for this article? The abstract does not reflect any innovation and not even quantitative results are not demonstrated by the authors. Please make clear the novelty and contribution of the manuscript and its results as compared to the extensive literature available.
ii. In line 95, C100 joist? Please shows the figure of the joist.
iii. In Fig. 12a, the value of y axis is not shown. Also, what does -1.4% means?
iv. In computational verification of stress state, the non-linear parameters of the concrete and rebar are not mentioned. How the crack and damage parameter are defined? Which Eos and material properties (Elasticity and plasticity) are used?
v. The paper does not provide a clear objective of the study. Considering the drawbacks of the numerical methods. How tolerance of results is considered in this text with the different number of calculations.
vi. The fcd parameters used in Fig 9, should be defined at first.
vii. Results are merely present and there are no scientific findings are discussed. What is the difference between this manuscript and the article https://doi.org/10.3390/ma15113770 that was published by the current authors?
viii. The English language shall be improved as well as the format of the manuscript.
Author Response

(The authors gave the same response as above.)

Round 2
Reviewer 2 Report
For buildings with a long history, it is necessary to take appropriate tests to ensure the safety of their continued use. After modification, the structure and content of the article are relatively complete, which has met the requirements of Materials.
Reviewer 3 Report
It can be accepted
Reviewer 4 Report
This time the paper has been fully improved and corresponding modifications have been conducted in all part of the article. In my opinion, the current version can be considered for publication.